environmental chemistry

carbon nanotubes, dispersion, sedimentation, aggregation, mixed surfactants

**Author for correspondence:**
Helian Li
e-mail: chm_lihl@ujn.edu.cn

This article has been edited by the Royal Society of Chemistry, including the commissioning, peer review process and editorial aspects up to the point of acceptance.

# Dispersion, sedimentation and aggregation of multi-walled carbon nanotubes as affected by single and binary mixed surfactants

## Helian Li and Yanhua Qiu

School of Water Conservancy and Environment, University of Jinan, Jinan 250022, People's Republic of China

HL, 0000-0003-3058-0109

Two commonly used dispersants, octyl phenol ethoxylate (Triton X-100) and sodium dodecyl sulfate (SDS), were employed to explore the effects of single or mixed surfactants on the dispersion, sedimentation and aggregation of multi-walled carbon nanotubes (MWCNTs). Non-ionic surfactant TX100 showed much superior capability to anionic surfactant SDS in dispersing MWCNTs due to the benzene ring structure in its tail group. The addition of SDS reduced the adsorption of TX100 on the surface of MWCNTs and the consequent suspension of MWCNTs. The dispersing ability of TX100–SDS binary mixture was between those of individual SDS and TX100. The introduction of SDS greatly retarded the sedimentation and aggregation of suspended MWCNTs. The critical coagulation concentration (CCC) values of suspended MWCNTs dispersed by TX100 (2000 mg l$^{-1}$), SDS (2000 mg l$^{-1}$) and TX100–SDS (2000 mg l$^{-1}$ of each component) were 48.6, 398 and 324 mM, respectively, for Na$^+$ treatments. The CCC values were much lower for Ca$^{2+}$ treatments, which were 30.4 and 32.1 mM, respectively, for MWCNTs dispersed by TX100 and TX100–SDS mixture. Overall, these results demonstrated that although the introduction of SDS did not improve the ability of TX100 in suspending MWCNTs, the suspensions exhibited more stable properties than those dispersed by TX100 alone. Our findings have important implications for the design of surfactant mixtures and the prediction of the behaviour and fate of MWCNTs in the water environment.

# 1. Introduction

Carbon nanotubes (CNTs) including single-walled carbon nanotubes (SWCNTs) and multi-walled carbon nanotubes (MWCNTs) are

promising nano-materials with remarkable electronic, mechanical, optical and thermal properties [1–3]. Nowadays, CNTs have been increasingly used in a wide variety of industrial applications and consumer products [4,5]. Due to strong van der Waals interactions along the length axis, they are prone to aggregation and formation of thick bundles, ropes and agglomerates [6]. Moreover, CNTs are highly hydrophobic and are poorly soluble in either water or organic solvents, which limit their application.

In most applications, CNTs can realize their extraordinary properties only if they are in a uniform and stable suspension state [6,7]. It has been well established that surfactants such as hexadecyl trimethyl ammonium bromide (CTAB) [8], octyl phenol ethoxylate (Triton X-100) [9], sodium dodecyl benzene sulfonate (SDBS) [10] and sodium dodecyl sulfate (SDS) [11,12] can be used to achieve stable CNT dispersions. The surfactants get adsorbed by way of their hydrophobic group onto the CNT exterior surface via non-covalent attraction forces [13] including hydrophobic interaction, hydrogen bonding, $\pi$–$\pi$ stacking and electrostatic interaction [8,14], which improve the dispersion of CNTs through steric or/and electrostatic repulsion [15]. There are several studies comparing the dispersion capabilities of different surfactants. For example, Rastogi et al. [9] found that the dispersing power of four surfactants follows the order of SDS < Tween 20 < Tween 80 < TX-100. Islam et al. [10] demonstrated that SDBS exhibited superior capability in dispersing SWCNTs than TX100 and SDS. Bai et al. [8] reported that the dispersion of MWCNTs in three surfactants followed the order of SDS < CTAB < TX100. The difference in the dispersion capabilities can be explained by graphite–surfactant interactions, alkyl chain length, headgroup size and surface charge density [10,16]. Owing to the fact that surfactant adsorption plays an essential role in the dispersion of CNTs, any change in the amount of the adsorption will unavoidably cause the variation of the dispersion. It is well known that surfactant adsorption is greatly affected by the mixed surfactant phenomenon [17]. In mixed surfactant systems, the critical micellar concentrations are lower than those of each component as a result of the non-ideal mixing, which will affect the distribution of surfactant molecules between monomer and micelle pseudophases and thus the adsorption of surfactant. In many industrial applications based on the adsorption of a surfactant such as thin-film formation [18], flotation [19] and wetting [20], the use of properly designed mixed surfactant systems offers some promising advantages over pure component systems. However, in the dispersion of CNTs, most of the literature deals with the effect of single surfactant systems. To the best of our knowledge, there are only a few studies to investigate the dispersive effects of the mixture of cationic surfactant and anionic surfactant, in which exceptionally stable CNT/graphene nanosheet dispersions were obtained using a mixture of dodecyl trimethyl ammonium bromide and sodium octanoate [21] or a mixture of CTAB and SDS [22,23] at much lower total surfactant concentration as compared with the concentration when used alone, reflecting a synergistic effect in these mixtures. Although the synergistic effects of mixed surfactant systems have been widely applied in improving soil washing/flushing efficiency and crude oil recovery [24,25], there are still gaps in the knowledge concerning the effects of mixed surfactants, especially the mixture of anionic and non-ionic surfactants, on the dispersion of CNTs.

Moreover, the extensive use of CNTs will inevitably result in their release to the aquatic environment. The surfactants used to disperse CNTs and those used in different industries and households will coexist in the water environment, imposing influence on the environmental behaviour of CNTs. How mixed surfactants affect the fate of CNTs in the water environment is largely unknown. A detailed understanding of this issue will not only improve the dispersion technology but also necessary for predicting the environmental behaviour and fate of MWCNTs in the water environment. Thus, the dispersion, sedimentation and aggregation of MWCNTs in the presence of single or mixed surfactants of SDS and TX100 were investigated in this study. SDS and TX100 were employed because they have been widely used to disperse nano-materials [9,11,12,26]. Moreover, the effects of their mixture on the dispersion, sedimentation and aggregation of MWCNTs still remain unknown. The aims were to (1) explore the dispersion of MWCNTs by SDS, TX100 and the binary mixed surfactant solutions (SDS–TX100) and (2) quantify and compare the sedimentation and aggregation kinetics of MWCNTs under the single and mixed surfactant treatments.

# 2. Material and methods

## 2.1. Materials

MWCNTs were purchased from Shenzhen Nanotech Co. (Shenzhen, China) with purity over 95% by weight and outer diameter of 60–100 nm. Detailed properties of the MWCNTs were reported by Lin & Xing [27] previously.

Sodium chloride (NaCl), calcium chloride dihydrate (CaCl$_2$·2H$_2$O) and SDS were obtained from Fisher Scientific. TX100 was provided by Acros (NJ, USA). The critical micellar concentrations of TX100 and SDS were 194 mg l$^{-1}$ [28] and 2307 mg l$^{-1}$ [16], respectively.

## 2.2. Dispersion of MWCNTs by single or mixed surfactants

Dispersion experiments were carried out with a batch equilibration method. About 40 ml of an aqueous solution with 0 or 2000 mg l$^{-1}$ TX100 and different concentrations of SDS were placed in 40 ml vials with 20 mg MWCNTs. The initial concentration of SDS was set as 0, 100, 200, 500, 1000 and 2000 mg l$^{-1}$ to explore its effect on TX100 adsorption and the subsequent dispersion of MWCNTs. The vials were sealed with Teflon screw caps and were shaken at 150 r.p.m. and 25°C for 7 days. The vials were then centrifuged at 3000 r.p.m. for 30 min, and an aliquot of the supernatant was collected for analysis of the dispersed MWCNTs. Another aliquot of the supernatant was collected and filtered through a 0.2 μm PTFE filter (Whatman) for analysis of the remaining TX100. The concentrations of MWCNTs and TX100 in the supernatant were determined using a UV–visible spectrometer (Agilent 8453, USA) at 800 nm and 223 nm, respectively. The hydrodynamic diameters ($D_h$) and zeta potentials of the supernatants were also measured using a 90 Plus Particle Size and Zeta Potential Analyzer (Brookhaven Instruments Corporation, NY, USA).

## 2.3. Sedimentation kinetics of MWCNTs dispersed by single or mixed surfactants

The MWCNT samples used for sedimentation experiments were prepared using an oscillation method. About 400 ml of surfactants at a concentration of 2000 mg l$^{-1}$ was added to 500 ml flasks with 200 mg MWCNTs. The flasks were sealed and shaken at 150 r.p.m. and 25°C for 7 days. Then, the flasks were left for settling at room temperature. At predetermined time intervals, an aliquot of the supernatants was carefully collected from just below the surface of the liquid for UV–visible analysis at 800 nm as described earlier. The ratios ($y$) of the absorbance at time $t$ to that at initial time were used to measure the stability of the suspensions. An exponential decay model (equation (2.1)) was used to fit the settling data of MWCNTs dispersed by different surfactants:

$$y = \mathrm{e}^{-kt}, \tag{2.1}$$

where $k$ is the rate constant of sedimentation (h$^{-1}$).

## 2.4. Aggregation kinetics of MWCNT suspension dispersed by single or mixed surfactants

The MWCNT samples used for aggregation experiments were prepared using a method similar to that of the sedimentation experiments. After 7 days of oscillation, the dispersions were kept at a standstill for 24 h; then, the supernatants were collected for aggregation experiments. The aggregation investigation was performed with dynamic light scattering (DLS). The detector employed a laser beam at 90° with a 15 s time interval. About 1.5 ml MWCNT stock suspension and a specific amount of each electrolyte (NaCl or CaCl$_2$) were pipetted into a DLS cuvette to initiate the aggregation. The initial concentration of MWCNTs was 7.4, 3.9 and 5.7 mg l$^{-1}$, respectively, for TX100, SDS and TX100–SDS treatment. The cuvette was vortexed for 1 s and then immediately placed in the DLS instrument. The $D_h$ measurements were conducted for a time period ranging from 10 to 60 min to obtain an approximately 30% increase in the original hydrodynamic radius of MWCNTs [29]. The initial aggregation rate constants ($k$) of MWCNTs are proportional to the initial rate of increase of $D_h$ with time and the inverse of MWCNT concentration $N_0$:

$$k \propto \frac{1}{N_0} \left( \frac{\mathrm{d}D_h(t)}{\mathrm{d}t} \right)_{t \to 0}. \tag{2.2}$$

As the MWCNT concentration in all aggregation experiments was identical, the attachment efficiency, $\alpha$, can be derived by normalizing the initial slope of the aggregation profile of a given solution chemistry by the initial slope obtained for favourable (fast) aggregation conditions:

$$\alpha = \frac{k}{k_{\mathrm{fast}}} = \frac{(\mathrm{d}D_h(t)/\mathrm{d}t)_{t \to 0}}{(\mathrm{d}D_h(t)/\mathrm{d}t)_{t \to 0,\mathrm{fast}}}, \tag{2.3}$$

where the subscript 'fast' represents the favourable (non-repulsive) aggregation conditions [30–32].

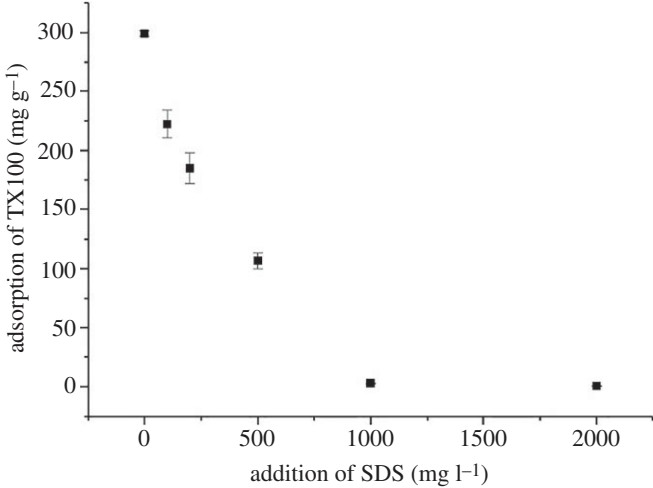

**Figure 1.** Adsorption of TX100 to the MWCNTs with the addition of SDS.

# 3. Results and discussion

## 3.1. Effect of SDS on the adsorption of TX100 to the surface of MWCNTs

Due to the fact that the dispersion of CNTs by a surfactant is a process based on adsorption, any influence on the adsorption of the surfactant would inevitably affect the dispersion of MWCNTs. The adsorption of surfactants in the binary mixed system would be greatly different from that in the single system. The adsorption of TX100 onto the surface of MWCNTs at different SDS concentrations is shown in figure 1. In the absence of SDS, the adsorption of TX100 on the MWCNTs was 299 mg g$^{-1}$, corresponding to 462 mmol kg$^{-1}$, which is lower than the value (607 mmol kg$^{-1}$) reported by Bai *et al.* [33]. The difference in sorption capacity is likely due to the different properties of the CNTs. The outer diameter of the CNTs in their study was less than 10 nm, having a much higher specific surface area than that of the MWCNTs used in this study. With the addition of SDS, the adsorption of TX100 decreased sharply (figure 1). In the presence of 2000 mg l$^{-1}$ of SDS, almost all the TX100 molecules were partitioned to the solution. This may be due to the following two reasons. First, there is a competition between SDS and TX100 for the adsorption sites on the surface of MWCNTs. Second, the addition of SDS resulted in the formation of mixed micelles at much lower critical micellar concentration than that of TX100 alone, which would compete with MWCNTs for the free TX100 molecules [34]. With the increased mass of SDS, the critical micellar concentration of the mixed surfactant system decreased, and more TX100 molecules participated in the formation of mixed micelles instead of being adsorbed onto the surface of MWCNTs. Since only surfactant monomers could be adsorbed by the MWCNTs, the adsorption of TX100 decreased as a result of the introduction of SDS. A similar influence of SDS on the sorption of TX100 onto soil surface has been reported by Zhou & Zhu [34]. The effect of SDS on the adsorption behaviour of TX100 would have a significant impact on the dispersion of MWCNTs, which will be discussed later.

## 3.2. Characteristics of MWCNT suspensions dispersed by single or mixed surfactants

The hydrodynamic diameters and zeta potentials of the MWCNT suspensions dispersed by different surfactants are shown in figure 2. The pH values of all the treatments were around 5.5. Moreover, all the MWCNT suspensions were negatively charged. The adsorption of anionic surfactant could transfer negative charges to the surface of MWCNTs, which made the zeta potentials more negative. Although the adsorption of non-ionic surfactant could not change the zeta potential, the MWCNT suspensions dispersed by TX100 were also negatively charged, which is attributed to the hydroxyl and carboxyl groups of MWCNTs that are created during strong acid treatment for end-cap opening and purification [27]. For SDS treatment, due to the transfer of negative charges from SDS to the surface of MWCNTs, the absolute values of zeta potential increased from 40 to 59 mV with the increase of SDS concentration from 100 to 2000 mg l$^{-1}$. The zeta potential of SDS treatments

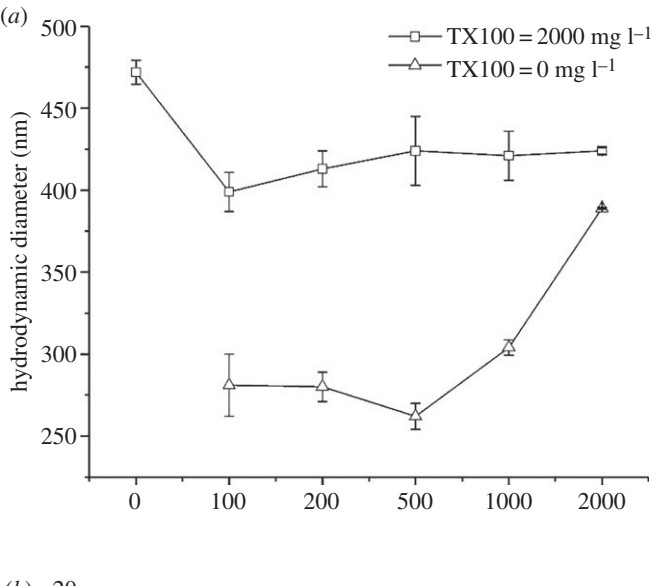

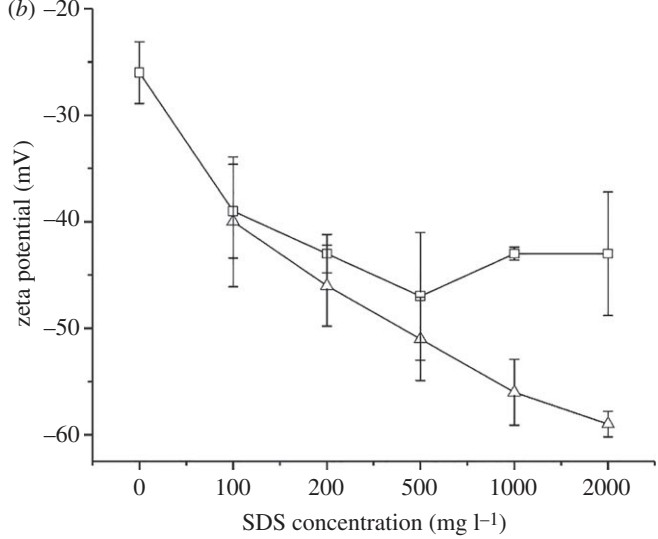

**Figure 2.** (*a*) Hydrodynamic diameters and (*b*) zeta potentials of MWCNT suspensions dispersed by single or mixed surfactants. Values are presented as mean ± s.d. ($n = 3$).

was greatly higher than those of TX100 and TX100–SDS treatments. At surfactant concentration of 2000 mg l$^{-1}$, the absolute value of zeta potential of SDS treatment is 59 mV, which is much higher than that of TX100 treatment (26 mV). For the TX100–SDS treatment, the absolute value of zeta potential of MWCNTs increased from 26 to 47 mV with the increase of SDS concentration from 0 to 500 mg l$^{-1}$, then decreased to 43 mV at SDS concentrations of 1000 and 2000 mg l$^{-1}$. The zeta potentials of TX100–SDS treatments were slightly lower than those of SDS treatments, indicating that the presence of TX100 resulted in charge screening effects. This is because the non-ionic hydrophilic group of TX100 shields the charged group on SDS.

With regard to $D_h$, the SDS treatments afforded much smaller values than those of TX100 and TX100–SDS treatments. In the SDS treatment, the $D_h$ decreased from 281 to 262 nm with SDS concentrations increased from 100 mg l$^{-1}$ to 500 mg l$^{-1}$, then increased to 389 nm at SDS concentration of 2000 mg l$^{-1}$. The increase in $D_h$ can be attributed to the bridging between CNTs by extra surfactants [9]. The $D_h$ of MWCNTs for 2000 mg l$^{-1}$ of SDS treatment was far below that of 2000 mg l$^{-1}$ of TX100 treatment (472 nm). This is presumably due to the smaller molecular size of SDS, which makes it more easily enter the narrow space between individual MWCNTs, leading to a smaller average hydrodynamic diameter of suspended MWCNTs. For the TX100–SDS treatments, with the introduction of 100 mg l$^{-1}$ of SDS, the $D_h$ of MWCNTs decreased sharply from 472 to 399 nm, then remained almost stable at around 420 nm despite the further increase in the SDS concentration.

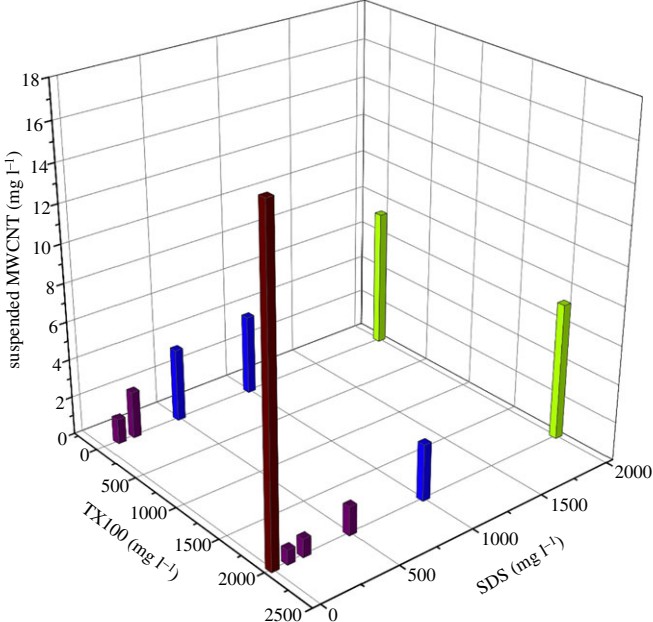

**Figure 3.** Suspension of MWCNTs by single or mixed surfactant solutions.

## 3.3. Concentrations of MWCNT suspensions dispersed by single or mixed surfactants

Figure 3 shows the suspended MWCNT concentrations dispersed by single or mixed surfactants. As can be seen, TX100 exhibited a much higher ability to disperse MWCNTs than SDS. At 2000 mg $l^{-1}$ of surfactant concentration, the suspended MWCNT concentration was 17.2 mg $l^{-1}$ for TX100 treatment, compared with 7.27 mg $l^{-1}$ for SDS treatment. This is due to the higher adsorption capacity of TX100 to the surface of MWCNTs compared with that of SDS. The benzene ring in the tail group of TX100 leads to $\pi$–$\pi$ stacking type interaction, increasing the binding and surface coverage of surfactant molecules to graphite significantly [10]. A former study has demonstrated that the benzene ring plays an essential role in the dispersion of MWCNTs using surfactants [9]. The addition of different concentrations of SDS greatly decreased the suspension of the MWCNTs. This can be explained by the decreased adsorption of TX100 in the presence of SDS (figure 1). Both SDS and TX100 could adsorb to the MWCNT surface through hydrophobic interaction. As a non-ionic surfactant, TX100 mainly disperses MWCNTs through steric hindrance mechanism. While for ionic surfactant SDS, both steric and electrostatic stabilization took effect in the dispersion of MWCNTs. In the TX100–SDS binary mixture system, the suspension of MWCNTs was the result of the comprehensive effect of decreased steric repulsion from TX100 and increased electrostatic stabilization from SDS. In both the single and mixed surfactant treatments, the suspended concentration of MWCNTs increased with increasing SDS concentration, which is presumably due to the increased electrostatic repulsion. As indicated in figure 2, the introduction of SDS greatly increased the absolute value of the zeta potential of MWCNTs. The addition of TX100 to SDS solution also decreased the suspension of MWCNTs compared with SDS treatment, which can be attributed to the decreased zeta potential in the presence of TX100 (figure 2), thus decreasing the repulsive forces between individual CNTs.

## 3.4. Effect of SDS on the sedimentation dynamics of MWCNT suspensions

Sedimentation experiments were carried out to investigate the stability of MWCNT suspensions dispersed by single or mixed surfactants. Figure 4 illustrates that MWCNTs dispersed by TX100 were less stable than those dispersed by SDS and TX100–SDS. Initially, the concentrations of suspended MWCNTs decreased sharply with increasing time in all the treatments. The MWCNTs dispersed by SDS and TX100–SDS even settled faster than those dispersed by TX100, with a rate constant of 0.002 compared to the value of 0.001 for TX100-dispersed MWCNTs (figure 4). After 600 h, the concentrations of suspensions dispersed by SDS and TX100–SDS became stable, while the suspension dispersed by TX100 still decreased with increasing time. MWCNTs dispersed by TX100 are more prone to sedimentation due to larger $D_h$ and lower zeta potential as compared to those dispersed by

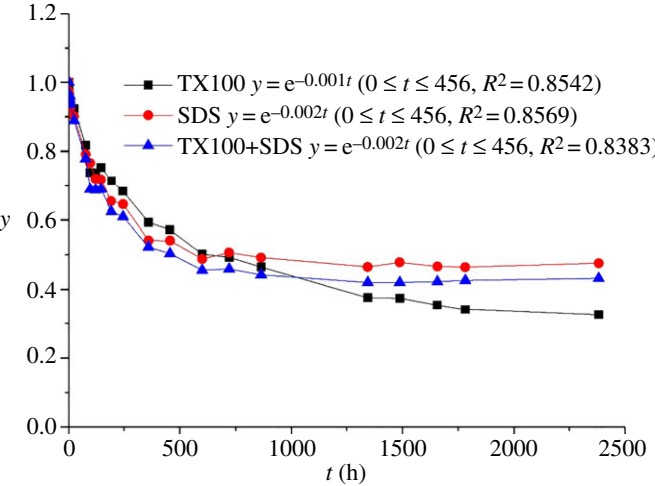

**Figure 4.** Sedimentation kinetics of MWCNT suspensions dispersed by single or mixed surfactants. The abscissa is the sedimentation time elapsed ($t$), $y$ indicates the ratio of the absorbance of MWCNT suspension at time $t$ to that at the initial time.

SDS (figure 2). At sedimentation equilibrium, the supernatant MWCNT concentrations were 49% and 46% of the initial values for SDS and TX100–SDS treatments, respectively. With regard to TX100 treatment, although the highest concentration of MWCNT suspension was obtained at the beginning, more than 60% of the dispersed MWCNTs were settled with increasing time. It is thus demonstrated that the introduction of SDS could effectively increase the stability of MWCNT suspensions. This is likely due to the significant increase in the zeta potential and decrease in $D_h$ as indicated in figure 2. Similar retardation of the sedimentation of nano-TiO$_2$ and nano-ZnO by SDS has been reported by Li *et al.* [35,36], in which they found that the repulsive force generated by the adsorbed SDS prevented the quick sedimentation of nanoparticles.

## 3.5. Effect of SDS on the aggregation behaviour of MWCNT suspensions

The attachment efficiencies of MWCNTs dispersed by single or mixed surfactants as functions of NaCl and CaCl$_2$ are plotted in figure 5. At pH around 5.5, the MWCNTs were negatively charged, thereby exhibiting strong electrostatic repulsion at low ionic strengths. Aggregation kinetics of the MWCNTs exhibited unfavourable (slow) and favourable (fast) regimes in the presence of both electrolytes, which indicated that Derjaguin–Landau–Verwey–Overbeek type interactions were the dominant mechanism of MWCNT stabilization. At low ionic strength, the attachment efficiency increased with increasing salt concentration. While at higher ionic strength, the attachment efficiency was not affected by the electrolyte concentration because the electrostatic repulsion between MWNTs was completely suppressed and every collision between MWNTs resulted in attachment [28]. The critical coagulation concentration (CCC) can be derived from the intersections between the extrapolations through the unfavourable and favourable regimes [29]. The CCC values of MWCNT suspensions dispersed by TX100, SDS and TX100–SDS mixture were estimated to be 48.6, 398 and 324 mM, respectively, for NaCl treatments. The CCC value for TX100-dispersed MWCNTs in the presence of NaCl was consistent with our former results despite the higher TX100 concentration used in this study [37]. In the presence of CaCl$_2$, the CCC values obtained were much lower, which were 30.4 and 32.1 mM, respectively, for MWCNTs dispersed by TX100 and TX100–SDS mixture. For TX100–SDS treatment, the CCC value in the presence of NaCl was greatly increased compared to that of TX100 treatment, while in the presence of CaCl$_2$, the change of CCC value was not significant due to the bridging effect of calcium ions. No CCC value was given for MWCNT suspension dispersed by SDS due to the formation of precipitation by SDS and Ca$^{2+}$. While in the presence of TX100, the salt tolerance of SDS was greatly increased, and no precipitation of SDS was observed in the Ca$^{2+}$ concentration range of this study. It is interesting to note that the CCC value of MWCNTs dispersed by TX100 in the presence of CaCl$_2$ is about six times that obtained in our former study, which is very different from that of the NaCl treatments [37]. It is suggested that the CCC values of Ca$^{2+}$ treatments were more sensitive to TX100 concentration changes, which needs to be studied further in the future. The

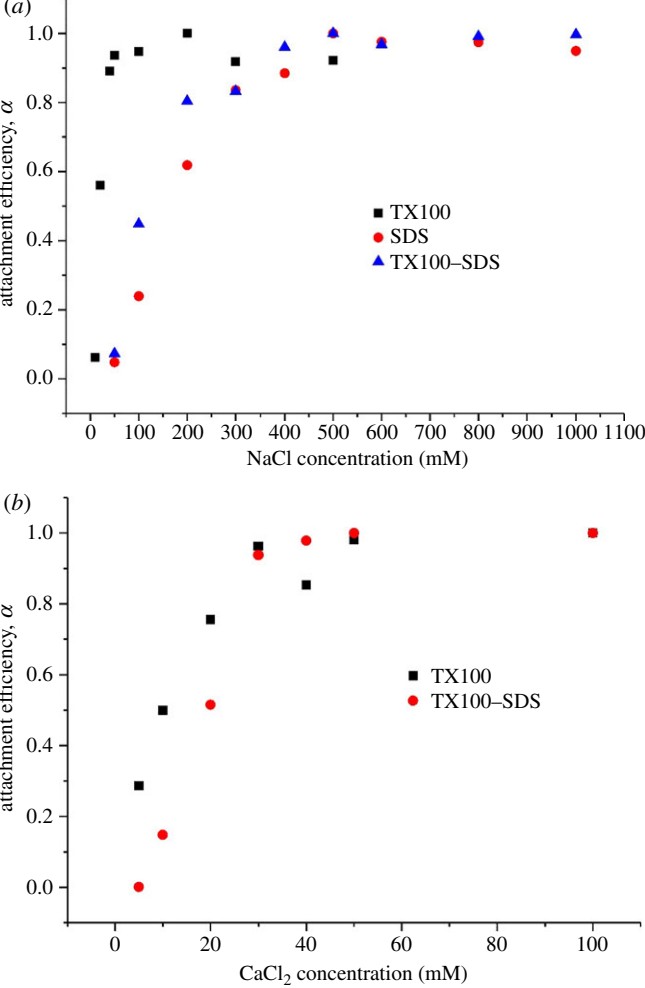

**Figure 5.** Attachment efficiencies of MWCNTs dispersed by single or mixed surfactants as functions of (*a*) NaCl and (*b*) CaCl$_2$.

addition of SDS led to two competing processes: stabilization of the nanoparticles due to the adsorption and aggregation of the nanoparticles due to the increase in ionic strength [38]. It is observed that 2000 mg l$^{-1}$ of SDS greatly increased the CCC values in NaCl treatments, indicating that the stabilization effect predominated in the effect of SDS. The introduction of SDS can drastically reduce the aggregation of MWCNT suspensions, which is largely due to the increase in zeta potential (figure 2).

## 4. Conclusion

TX100 showed much higher ability in dispersing MWCNTs than SDS, while SDS-dispersed MWCNTs are more stable than those dispersed by TX100 or TX100–SDS mixture. The introduction of SDS into TX100 solution greatly decreased the adsorption of TX100 to MWCNTs, thus compromising the ability to suspend MWCNTs. However, the sedimentation and aggregation of MWCNT suspensions were greatly retarded in the presence of SDS, resulting in the improvement of the stability of the MWCNT suspensions. Our findings suggest that the properties and environmental behaviour of MWCNTs would be quite different in the single and mixed surfactant systems, which should be properly addressed in predicting the fate of MWCNTs in the water environment.

Data accessibility. Data are available from the Dryad Digital Repository: https:// doi.org/10.5061/dryad.v19s04p [39].
Authors' contributions. H.L. conceptualized the study, analysed the data and prepared the manuscript. Y.Q. performed the experiments and analysed the data.
Competing interests. The authors have no competing interests.
Funding. This work was supported by the National Natural Science Foundation of China (grant number 41771524).
Acknowledgements. The authors thank Prof. Baoshan Xing from the University of Massachusetts Amherst for providing the multi-walled carbon nanotubes.

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
