## [Reviewer comments · Royal Society Open Science]

Review History

RSOS-190241.R0 (Original submission)

Review form: Reviewer 1

Is the manuscript scientifically sound in its present form?

No

Are the interpretations and conclusions justified by the results?

Yes

Is the language acceptable?

Yes

Is it clear how to access all supporting data?

Not Applicable

Do you have any ethical concerns with this paper?

No

Have you any concerns about statistical analyses in this paper?

No

Recommendation?

Reject

Comments to the Author(s)

The work presented by Li et al. reports on the dispersibility properties of MWCNTs in two very common surfactants. In reported literature is possible to find very similar publications since more than 15 years up to now. I can not see any new information or contribution in the experimental protocol, characterization techniques or new results. Royal Society Open Science is an interdisciplinary journal. The present work is a routine protocol for the processing of MWCNTs, thus I do not suggest its publication.

Review form: Reviewer 2

Is the manuscript scientifically sound in its present form?

Yes

Are the interpretations and conclusions justified by the results?

Yes

Is the language acceptable?

Yes

Is it clear how to access all supporting data?

Yes

Do you have any ethical concerns with this paper?

No

Have you any concerns about statistical analyses in this paper?

No

Recommendation?

Accept with minor revision (please list in comments)

Comments to the Author(s)

The novel nano-carbon based materials such as CNT, C60 and GR et. al show great application potential in the modern life. The improvements of material performance and the safety application in practice are both significantly important for the promising nanomaterials. In this study, Li and his co-authors systematic investigated the dispersion, sedimentation and aggregation of the multi-walled carbon nanotubes under conditions of single and binary mixed surfactants in aqueous solution. Overall the approach is not novel, the results are very interesting and do contribute to new scientific knowledge on the design of surfactant mixtures and the prediction on the behavior and fate of MWCNTs in the water environment. What's more, the organization and writing of the manuscript is well. After a through reading of this manuscript, I think this study is sound with only some comments that need to be addressed before this could be considered for publication in Royal Society Open Science.

Special comments:

1. There are some grammatical and writing errors, which should be checked throughout the text, such as line 71, line 75, line 147 etc.
2. In this study, for what's reason the authors select the surfactants TX-100 and SDS as the representative surfactant, nor the surfactants such as SDBS, CTAB and Tween 80 et al, which should be state clearly in the manuscript.
3. In page 9, line 175-181, the authors have described the variation tendency of the Dh with the different addition of SDS concentration, and the results show two different properties of MWCNT suspensions in the single and mixed surfactants. I suggest the authors explain the different dispersed phenomenon form the basic structure and properties of the two surfactants in the manuscript.
4. In page 12, Figure 3, I suggest the authors compare the sedimentation kinetics of MWCNT in the condition of bare TX-100 and SDS as surfactant in the manuscript, which can combine with the data of Dh and ξ -Potential in Table 1.
5. In page 13, line 239~244, why the CCC values for the MWCNT dispersed in TX100 and TX100-SDS show little difference under the condition of CaCl₂, while the CCC values have a distinct change in the NaCl treatment.
6. The style of reference should be uniform in the manuscript, some writing errors should be revised, eg. line 325, line 341 etc. In addition, the reference's DOI also need to provide in the manuscript to ensure that it conforms to the standard of the journal.

Review form: Reviewer 3

Is the manuscript scientifically sound in its present form?

Yes

Are the interpretations and conclusions justified by the results?

Yes

Is the language acceptable?

Yes

Is it clear how to access all supporting data?

Not Applicable

Do you have any ethical concerns with this paper?

No

Have you any concerns about statistical analyses in this paper?

No

Recommendation?

Accept with minor revision (please list in comments)

Comments to the Author(s)

The manuscript deals with carbon nanotubes dispersion in surfactant systems. Although there are a number of works on this aspect, the topic is interesting and there are new aspects worth of being published. Introduction is not up-to-date. Recent references on micro-sized nanotubes stabilization by surfactant systems have to be added (see for instance: Ceramics International 2019, 45 (2, Part B), 2858–2865. <https://doi.org/10.1016/j.ceramint.2018.07.289>.; Langmuir 2015, 31 (27), 7472–7478. <https://doi.org/10.1021/acs.langmuir.5b01181>).

Other comments are reported below.

- Figure 2. Labels are not clear and readable. Authors should revise.
- Data in table 1 should be better presented in a figure for a quick visualization.
- Sedimentation. The discussion might be improved by fitting the experimental sedimentation data by proper kinetic model. Authors should try in this direction.

Decision letter (RSOS-190241.R0)

24-Apr-2019

Dear Dr Li:

Title: Dispersion, sedimentation, and aggregation of multi-walled carbon nanotubes as affected by single and binary mixed surfactants
Manuscript ID: RSOS-190241

The editor assigned to your manuscript has now received comments from reviewers. We would like you to revise your paper in accordance with the referee and Subject Editor suggestions which can be found below (not including confidential reports to the Editor). Please note this decision does not guarantee eventual acceptance.

Please submit your revised paper before 17-May-2019. Please note that the revision deadline will expire at 00.00am on this date. If we do not hear from you within this time then it will be assumed that the paper has been withdrawn. In exceptional circumstances, extensions may be possible if agreed with the Editorial Office in advance. We do not allow multiple rounds of revision so we urge you to make every effort to fully address all of the comments at this stage. If deemed necessary by the Editors, your manuscript will be sent back to one or more of the original reviewers for assessment. If the original reviewers are not available we may invite new reviewers.

Please also include the following statements alongside the other end statements. As we cannot publish your manuscript without these end statements included, if you feel that a given heading is not relevant to your paper, please nevertheless include the heading and explicitly state that it is not relevant to your work.

- Ethics statement

Please clarify whether you received ethical approval from a local ethics committee to carry out your study. If so please include details of this, including the name of the committee that gave

consent in a Research Ethics section after your main text. Please also clarify whether you received informed consent for the participants to participate in the study and state this in your Research Ethics section.

OR

Please clarify whether you obtained the necessary licences and approvals from your institutional animal ethics committee before conducting your research. Please provide details of these licences and approvals in an Animal Ethics section after your main text.

OR

Please clarify whether you obtained the appropriate permissions and licences to conduct the fieldwork detailed in your study. Please provide details of these in your methods section.

On behalf of the Subject Editor Professor Anthony Stace and the Associate Editor Professor Claire Carmalt.

RSC Associate Editor:

Comments to the Author:

Although Reviewer 1 did not support publication of your manuscript, because Reviewers 2 and 3 supported the publication of the work after revisions, I would like to recommend your manuscript for revisions.

RSC Subject Editor:

Comments to the Author:

(There are no comments.)

Reviewers' Comments to Author:

Reviewer: 1

Comments to the Author(s)

The work presented by Li et al. reports on the dispersibility properties of MWCNTs in two very common surfactants. In reported literature is possible to find very similar publications since more than 15 years up to now. I can not see any new information or contribution in the experimental protocol, characterization techniques or new results. Royal Society Open Science is an interdisciplinary journal. The present work is a routine protocol for the processing of MWCNTs, thus I do not suggest its publication.

Reviewer: 2

Comments to the Author(s)

The novel nano-carbon based materials such as CNT, C60 and GR et. al show great application potential in the modern life. The improvements of material performance and the safety application in practice are both significantly important for the promising nanomaterials. In this study, Li and his co-authors systematic investigated the dispersion, sedimentation and aggregation of the multi-walled carbon nanotubes under conditions of single and binary mixed surfactants in aqueous solution. Overall the approach is not novel, the results are very interesting and do contribute to new scientific knowledge on the design of surfactant mixtures and the prediction on the behavior and fate of MWCNTs in the water environment. What's more, the organization and writing of the manuscript is well. After a through reading of this manuscript, I think this study is sound with only some comments that need to be addressed before this could be considered for publication in Royal Society Open Science.

Special comments:

1. There are some grammatical and writing errors, which should be checked throughout the text, such as line 71, line 75, line 147 etc.
2. In this study, for what's reason the authors select the surfactants TX-100 and SDS as the representative surfactant, nor the surfactants such as SDBS, CTAB and Tween 80 et al, which should be state clearly in the manuscript.
3. In page 9, line 175-181, the authors have described the variation tendency of the D_h with the different addition of SDS concentration, and the results show two different properties of MWCNT suspensions in the single and mixed surfactants. I suggest the authors explain the different dispersed phenomenon form the basic structure and properties of the two surfactants in the manuscript.
4. In page 12, Figure 3, I suggest the authors compare the sedimentation kinetics of MWCNT in the condition of bare TX-100 and SDS as surfactant in the manuscript, which can combine with the data of D_h and ξ -Potential in Table 1.
5. In page 13, line 239~244, why the CCC values for the MWCNT dispersed in TX100 and TX100-SDS show little difference under the condition of CaCl₂, while the CCC values have a distinct change in the NaCl treatment.
6. The style of reference should be uniform in the manuscript, some writing errors should be revised, eg. line 325, line 341 etc. In addition, the reference's DOI also need to provide in the manuscript to ensure that it conforms to the standard of the journal.

Reviewer: 3

Comments to the Author(s)

The manuscript deals with carbon nanotubes dispersion in surfactant systems. Although there are a number of works on this aspect, the topic is interesting and there are new aspects worth of being published. Introduction is not up-to-date. Recent references on micro-sized nanotubes stabilization by surfactant systems have to be added (see for instance: *Ceramics International* 2019, 45 (2, Part B), 2858–2865. <https://doi.org/10.1016/j.ceramint.2018.07.289>.; *Langmuir* 2015, 31 (27), 7472–7478. <https://doi.org/10.1021/acs.langmuir.5b01181>).

Other comments are reported below.

- Figure 2. Labels are not clear and readable. Authors should revise.
- Data in table 1 should be better presented in a figure for a quick visualization.
- Sedimentation. The discussion might be improved by fitting the experimental sedimentation data by proper kinetic model. Authors should try in this direction.

Author's Response to Decision Letter for (RSOS-190241.R0)

See Appendix A.

RSOS-190241.R1 (Revision)

Review form: Reviewer 2

Is the manuscript scientifically sound in its present form?

Yes

Are the interpretations and conclusions justified by the results?

Yes

Is the language acceptable?

Yes

Do you have any ethical concerns with this paper?

No

Recommendation?

Accept as is

Comments to the Author(s)

Thanks for the authors hard working on response all the comments of reviewers point by point. I think this work can be acceptable on the journal of "Royal society Open Science".

Decision letter (RSOS-190241.R1)

25-Jun-2019

Dear Dr Li:

Title: Dispersion, sedimentation, and aggregation of multi-walled carbon nanotubes as affected by single and binary mixed surfactants
Manuscript ID: RSOS-190241.R1

It is a pleasure to accept your manuscript in its current form for publication in Royal Society Open Science. The chemistry content of Royal Society Open Science is published in collaboration with the Royal Society of Chemistry.

RSC Associate Editor:
Comments to the Author:
(There are no comments.)

RSC Subject Editor:
Comments to the Author:
(There are no comments.)

Reviewer(s)' Comments to Author:
Reviewer: 2

Comments to the Author(s)

Thanks for the authors hard working on response all the comments of reviewers point by point. I think this work can be acceptable on the journal of "Royal society Open Science".

Appendix A

Reviewer: 1

Comments to the Author(s)

The work presented by Li et al. reports on the dispersibility properties of MWCNTs in two very common surfactants. In reported literature is possible to find very similar publications since more than 15 years up to now. I can not see any new information or contribution in the experimental protocol, characterization techniques or new results. Royal Society Open Science is an interdisciplinary journal. The present work is a routine protocol for the processing of MWCNTs, thus I do not suggest its publication.

Thanks. Although there are a lot of publications on the dispersion of MWCNTs by single surfactant, how the mixture of these surfactants on the behaviors of MWCNTs in water environment still remains unknown. We believe that our results are helpful on the design of surfactant mixtures and the prediction on the behavior and fate of MWCNTs in the water environment.

Reviewer: 2

We greatly appreciate your helpful comments on our paper. We have carefully revised our paper according to your comments. Please see responses in italicized font.

Comments to the Author(s)

The novel nano-carbon based materials such as CNT, C60 and GR et. al show great application potential in the modern life. The improvements of material performance and the safety application in practice are both significantly important for the promising nanomaterials. In this study, Li and his co-authors systematic investigated the dispersion, sedimentation and aggregation of the multi-walled carbon nanotubes under conditions of single and binary mixed surfactants in aqueous solution. Overall the approach is not novel, the results are very interesting and do contribute to new scientific knowledge on the design of surfactant mixtures and the prediction on the behavior and fate of MWCNTs in the water environment. What's more, the organization and writing of the manuscript is well. After a through reading of this manuscript, I think this study is sound with only some comments that need to be addressed before this could be considered for publication in Royal Society Open Science.

Special comments:

1. There are some grammatical and writing errors, which should be checked throughout the text, such as line 71, line 75, line 147 etc.

Thanks. We checked the whole manuscript and corrected the grammatical and writing errors carefully. Please see P. 3, line 62; P. 4, line 70; P. 7, line 152.

2. In this study, for what's reason the authors select the surfactants TX-100 and SDS as the representative surfactant, nor the surfactants such as SDBS, CTAB and Tween 80 et al, which should be state clearly in the manuscript.

SDS and TX100 were employed because they had been widely used to disperse nano-materials. Moreover, the effects of their mixture on the dispersion, sedimentation, and aggregation of MWCNTs still remains unknown. Please see P. 4, lines 73-76.

3. In page 9, line 175-181, the authors have described the variation tendency of the D_h with the different addition of SDS concentration, and the results show two different properties of MWCNT suspensions in the single and mixed surfactants. I suggest the authors explain the different dispersed phenomenon form the basic structure and properties of the two surfactants in the manuscript.

We try to explain the difference in the D_h from the molecular size of SDS and TX100, please see P. 9-10, lines 186-188. Moreover, the increase of D_h with increasing SDS concentration was explained in the manuscript. Please see P. 9, lines 184-185.

4. In page 12, Figure 3, I suggest the authors compare the sedimentation kinetics of MWCNT in the condition of bare TX-100 and SDS as surfactant in the manuscript, which can combine with the data of D_h and ξ -Potential in Table 1.

The comparison on the sedimentation kinetics of MWCNTs dispersed by bare TX 100 and SDS were conducted. Please see P. 12, lines 226-228.

5. In page 13, line 239~244, why the CCC values for the MWCNT dispersed in TX100 and TX100-SDS show little difference under the condition of CaCl_2 , while the CCC values have a distinct change in the NaCl treatment.

The insignificant change of CCC in the presence of CaCl_2 can be attributed to the bridging effect of calcium ions. Please see P. 13, lines 258-261.

6. The style of reference should be uniform in the manuscript, some writing errors should be revised, eg. line 325, line 341 etc. In addition, the reference's DOI also need to provide in the manuscript to ensure that it conforms to the standard of the journal.

Thanks. We corrected the errors in the references and added the DOI numbers. Please see P. 16-19, lines 299-393.

Reviewer: 3

Thank you so much for your comments. We read the comments carefully and revised our manuscript accordingly.

Comments to the Author(s)

The manuscript deals with carbon nanotubes dispersion in

surfactant systems. Although there are a number of works on this aspect, the topic is interesting and there are new aspects worth of being published. Introduction is not up-to-date. Recent references on micro-sized nanotubes stabilization by surfactant systems have to be added (see for instance: Ceramics International 2019, 45 (2, Part B), 2858–2865. <https://doi.org/10.1016/j.ceramint.2018.07.289>.; Langmuir 2015, 31 (27), 7472–7478. <https://doi.org/10.1021/acs.langmuir.5b01181>).

Thanks for the references you recommended to us. We carefully read the references and cited in the manuscript. Please see P. 4, line 74.

Other comments are reported below.

- Figure 2. Labels are not clear and readable. Authors should revise.

We revised the labels to make it more clear. Please see P. 11, Figure 3.

- Data in table 1 should be better presented in a figure for a quick visualization.

Data in table 1 were presented in a figure (P. 9, Figure 2) in this revised manuscript.

- Sedimentation. The discussion might be improved by fitting the experimental sedimentation data by proper kinetic model. Authors

should try in this direction.

An exponential decay model was used to fit the settling data from 0 to 476 h (nearly at sedimentation equilibrium). Please see P. 5-6, lines 107-110; P. 11-12, lines 222-224; P. 12, Figure 3.